# Multimodal Model-Agnostic Meta-Learning via Task-Aware Modulation

**Risto Vuorio**[*1]     **Shao-Hua Sun**[*2]     **Hexiang Hu**[2]     **Joseph J. Lim**[2]
[1]University of Michigan     [2]University of Southern California
vuoristo@gmail.com     {shaohuas, hexiangh, limjj}@usc.edu

## Abstract

Model-agnostic meta-learners aim to acquire meta-learned parameters from similar tasks to adapt to novel tasks from the same distribution with few gradient updates. With the flexibility in the choice of models, those frameworks demonstrate appealing performance on a variety of domains such as few-shot image classification and reinforcement learning. However, one important limitation of such frameworks is that they seek a common initialization shared across the entire task distribution, substantially limiting the diversity of the task distributions that they are able to learn from. In this paper, we augment MAML [5] with the capability to identify the mode of tasks sampled from a multimodal task distribution and adapt quickly through gradient updates. Specifically, we propose a multimodal MAML (MMAML) framework, which is able to modulate its meta-learned prior parameters according to the identified mode, allowing more efficient fast adaptation. We evaluate the proposed model on a diverse set of few-shot learning tasks, including regression, image classification, and reinforcement learning. The results not only demonstrate the effectiveness of our model in modulating the meta-learned prior in response to the characteristics of tasks but also show that training on a multimodal distribution can produce an improvement over unimodal training. The code for this project is publicly available at https://vuoristo.github.io/MMAML.

## 1   Introduction

Humans make effective use of prior knowledge to acquire new skills rapidly. When the skill of interest is related to a wide range of skills that one have mastered before, we can recall relevant knowledge of prior skills and exploit them to accelerate the new skill acquisition procedure. For example, imagine that we are learning a novel snowboarding trick with knowledge of basic skills about snowboarding, skiing, and skateboarding. We accomplish this feat quickly by exploiting our basic snowboarding knowledge together with inspiration from our skiing and skateboarding experience.

Can machines likewise quickly master a novel skill based on a variety of related skills they have already acquired? Recent advances in meta-learning [48, 6, 4] have attempted to tackle this problem. They offer machines a way to rapidly adapt to a new task using few samples by first learning an internal representation that matches similar tasks. Such representations can be learned by considering a distribution over similar tasks as the training data distribution. Model-based (*i.e.* RNN-based) meta-learning approaches [4, 52, 27, 25] propose to recognize the task identity from a few sample data, use the task identity to adjust a model's state (*e.g.* RNN's internal state or an external memory) and make the appropriate predictions with the adjusted model. Those methods demonstrate good performance at the expense of having to hand-design architectures, yet the optimal strategy of designing a meta-learner for arbitrary tasks may not always be obvious to humans. On the other hand, model-agnostic meta-learning frameworks [5, 7, 15, 18, 8, 28, 36, 35] seek an initialization of model

---

[*]Contributed equally.

parameters that a small number of gradient updates will lead to superior performance on a new task. With the flexibility in the model choices, these frameworks demonstrate appealing performance on a variety of domains, including regression, image classification, and reinforcement learning.

While most of the existing model-agnostic meta-learners rely on a single initialization, different tasks sampled from a complex task distributions can require substantially different parameters, making it difficult to find a single initialization that is close to all target parameters. If the task distribution is multimodal with disjoint and far apart modes (*e.g.* snowboarding, skiing), one can imagine that a set of separate meta-learners with each covering one mode could better master the full distribution. However, associating each task with one of the meta-learners not only requires additional task identity information, which is often not available or could be ambiguous when the modes are not clearly disjoint, but also disables transferring knowledge across different modes of the task distribution. To overcome this issue, we aim to develop a meta-learner that is able to acquire mode-specific prior parameters and adapt quickly given tasks sampled from a multimodal task distribution.

To this end, we leverage the strengths of the two main lines of existing meta-learning techniques: model-based and model-agnostic meta-learning. Specifically, we propose to augment MAML [5] with the capability of generalizing across a multimodal task distribution. Instead of learning a single initialization point in the parameter space, we propose to first compute the task identity of a sampled task by examining task related data samples. Given the estimated task identity, our model then performs modulation to condition the meta-learned initialization on the inferred task mode. Then, with these modulated parameters as the initialization, a few steps of gradient-based adaptation are performed towards the target task to progressively improve its performance. An illustration of our proposed framework is shown in Figure 1.

To investigate whether our method can acquire meta-learned prior parameters by learning tasks sampled from multimodal task distributions, we design and conduct experiments on a variety of domains, including regression, image classification, and reinforcement learning. The results demonstrate the effectiveness of our approach against other systems. A further analysis has also shown that our method learns to identify task modes without extra supervision.

The main contributions of this paper are three-fold as follows:

- We identify and empirically demonstrate the limitation of having to rely on a single initialization in a family of widely used model-agnostic meta-learners.
- We propose a framework together with an algorithm to address this limitation. Specifically, it generates a set of meta-learned prior parameters and adapts quickly given tasks from a multimodal task distribution leveraging both model-based and model-agnostic meta-learning.
- We design a set of multimodal meta-learning problems and demonstrate that our model offers a better generalization ability in a variety of domains, including regression, image classification, and reinforcement learning.

## 2   Related Work

The idea of empowering the machines with the capability of *learning to learn* [44] has been widely explored by the machine learning community. To improve the efficiency of handcrafted optimizers, a flurry of recent works has focused on learning to optimize a learner model. Pioneered by [38, 2], optimization algorithms with learned parameters have been proposed, enabling the automatic exploitation of the structure of learning problems. From a reinforcement learning perspective, [21] represents an optimization algorithm as a learning policy. [1] trains LSTM optimizers to learn update rules from the gradient history, and [34] trains a meta-learner LSTM to update a learner's parameters. Similar approach for continual learning is explored in [49].

Recently, investigating how we can replicate the ability of humans to learn new concepts from one or a few instances, known as *few-shot learning*, has drawn people's attention due to its broad applicability to different fields. To classify images with few examples, metric-based meta-learning frameworks have been proposed [16, 48, 42, 41, 43, 29, 3], which strive to learn a metric or distance function that can be used to compare two different samples effectively. Recent works along this line [29, 53, 19] share a conceptually similar idea with us and seek to perform task-specific adaptation with different type transformations. Due to the limited space, we defer the detailed discussion to the supplementary

material. While impressive results have been shown, it is nontrivial to adopt them for complex tasks such as acquiring robotic skills using reinforcement learning [12, 22, 14, 33, 9, 10, 20].

On the other hand, instead of learning a metric, model-based (*i.e.* RNN-based) meta-learning models learn to adjust model states (*e.g.* a state of an RNN [25, 4, 51] or external memory [37, 27]) using a training dataset and output the parameters of a learned model or the predictions given test inputs. While these methods have the capacity to learn any mapping from datasets and test samples to their labels, they could suffer from overfitting and show limited generalization ability [6].

Model-agnostic meta-learners [5, 7, 15, 18, 8, 28, 36, 35] are agnostic to concrete model configurations. Specifically, they aim to learn a parameter initialization under a certain task distribution, that aims to provide a favorable inductive bias for fast gradient-based adaptation. With its model agnostic nature, appealing results have been shown on a variety of learning problems. However, assuming tasks are sampled from a concentrated distribution and pursuing a common initialization to all tasks can substantially limit the performance of such methods on multimodal task distributions where the center in the task space becomes ambiguous.

In this paper, we aim to develop a more powerful model-agnostic meta-learning framework which is able to deal with complex multimodal task distributions. To this end, we propose a framework, which first identifies the mode of sampled tasks, similar to model-based meta-learning approaches, and then it modulates the meta-learned prior parameters to make the model better fit to the identified mode. Finally, the model is fine-tuned on the target task rapidly through gradient steps.

# 3   Preliminaries

The goal of meta-learning is to quickly learn task-specific functions that map between input data and the desired output $(x_k, y_k)_{k=1}^{K_t}$ for different tasks $t$, where the number of data $K_t$ is small. A task is defined by the underlying data generating distribution $\mathcal{P}(X)$ and a conditional probability $\mathcal{P}_t(Y \mid X)$. For instance, we consider five-way image classification tasks with $x_k$ to be images and $y_k$ to be the corresponding labels, sampled from a task distribution. The data generating distribution is unimodal if it contains classification tasks that belong to a single input and label domain (*e.g.* classifying different combination of digits). A multimodal counterpart therefore contains classification tasks from multiple different input and label domains (*e.g.* classifying digits vs. classifying birds). We denote the later distribution of tasks to be the *multimodal task distribution*.

In this paper, we aim to rapidly adapt to a novel task sampled from a multimodal task distribution. We consider a target dataset $\mathcal{D}$ consisting of tasks sampled from a multimodal distribution. The dataset is split into meta-training and meta-testing sets, which are further divided into task-specific training $\mathcal{D}_{\mathcal{T}}^{\text{train}}$ and validation $\mathcal{D}_{\mathcal{T}}^{\text{val}}$ sets. A meta-learner learns about the underlying structure of the task distribution through training on the meta-training set and is evaluated on meta-testing set.

Our work builds upon Model-Agnostic Meta-Learning (MAML) algorithm [5]. MAML seeks an initialization of parameters $\theta$ for a meta-learner such that it can be optimized towards a new task with a small number of gradient steps minimizing the task-specific objectives on the training data $\mathcal{D}_{\mathcal{T}}^{\text{train}}$, with the adapted parameters generalize well to the validation data $\mathcal{D}_{\mathcal{T}}^{\text{val}}$. The initialization of the parameters is trained by sampling mini-batches of tasks from $\mathcal{D}$, computing the adapted parameters for all $\mathcal{D}_{\mathcal{T}}^{\text{train}}$ in the batch, evaluating adapted parameters to compute the validation losses on the $\mathcal{D}_{\mathcal{T}}^{\text{val}}$ and finally update the initial parameters $\theta$ using the gradients from the validation losses.

# 4   Method

Our goal is to develop a framework to quickly master a novel task from *a multimodal task distribution*. We call the proposed framework Multimodal Model-Agnostic Meta-Learning (MMAML). The main idea of MMAML is to leverage two complementary neural networks to quickly adapt to a novel task. First, a network called the modulation network predicts the identity of the mode of a task. Then the predicted mode identity is used as an input by a second network called the task network, which is further adapted to the task using gradient-based optimization. Specifically, the modulation network accesses data points from the target task and produces a set of task-specific parameters to modulate the meta-learned prior parameters of the task network. Finally, the modulated task network (but

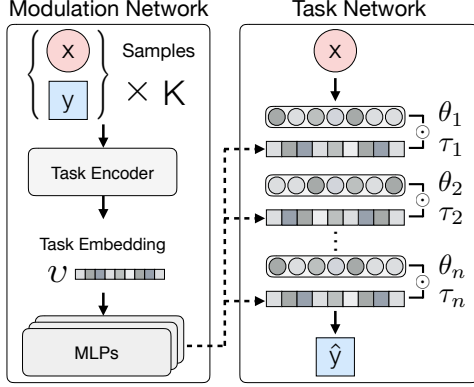

Modulation Network　　Task Network

**Algorithm 1** MMAML META-TRAINING PROCEDURE.

1: **Input:** Task distribution $P(\mathcal{T})$, Hyper-parameters $\alpha$ and $\beta$
2: Randomly initialize $\theta$ and $\omega$.
3: **while** not DONE **do**
4:　　Sample batches of tasks $\mathcal{T}_j \sim P(\mathcal{T})$
5:　　**for all** j **do**
6:　　　Infer $v = h(\{x, y\}_K; \omega_h)$ with K samples from $\mathcal{D}^{\mathrm{train}}_{\mathcal{T}_j}$.
7:　　　Generate parameters $\tau = \{g_i(v; \omega_g) \mid i = 1, \cdots, N\}$
　　　　to modulate each block of the task network $f$.
8:　　　Evaluate $\nabla_\theta \mathcal{L}_{\mathcal{T}_j}(f(x; \theta, \tau); \mathcal{D}^{\mathrm{train}}_{\mathcal{T}_j})$ w.r.t the K samples
9:　　　Compute adapted parameter with gradient descent:
　　　　$\theta'_{\mathcal{T}_j} = \theta - \alpha \nabla_\theta \mathcal{L}_{\mathcal{T}_j}(f(x; \theta, \tau); \mathcal{D}^{\mathrm{train}}_{\mathcal{T}_j})$
10:　　**end for**
11:　　Update $\theta$ with $\beta \nabla_\theta \sum_{T_j \sim P(\mathcal{T})} \mathcal{L}_{\mathcal{T}_j}(f(x; \theta', \tau); \mathcal{D}^{\mathrm{val}}_{\mathcal{T}_j})$
12:　　Update $\omega_g$ with $\beta \nabla_{\omega_g} \sum_{T_j \sim P(\mathcal{T})} \mathcal{L}_{\mathcal{T}_j}(f(x; \theta', \tau); \mathcal{D}^{\mathrm{val}}_{\mathcal{T}_j})$
13:　　Update $\omega_h$ with $\beta \nabla_{\omega_h} \sum_{T_j \sim P(\mathcal{T})} \mathcal{L}_{\mathcal{T}_j}(f(x; \theta', \tau); \mathcal{D}^{\mathrm{val}}_{\mathcal{T}_j})$
14: **end while**

Figure 1: **Model overview.** The modulation network produces a task embedding $v$, which is used to generate parameters $\{\tau_i\}$ that modulates the task network. The task network adapts modulated parameters to fit to the target task.

not the task-specific parameters from modulation network) is further adapted to target task through gradient-based optimization. A conceptual illustration can be found in Figure 1.

In the rest of this section, we introduce our modulation network and a variety of modulation operators in section 4.1. Then we describe our task network and the training details for MMAML in section 4.2.

## 4.1 Modulation Network

As mentioned above, modulation network is responsible for identifying the mode of a sampled task, and generate a set of parameters specific to the task. To achieve this, it first takes the given $K$ data points and their labels $\{x_k, y_k\}_{k=1,\ldots,K}$ as input to the task encoder $f$ and produces an embedding vector $v$ that encodes the characteristics of a task:

$$v = h\Big(\{(x_k, y_k) \mid k = 1, \cdots, K\}; \ \omega_h\Big) \tag{1}$$

Then the task-specific parameters $\tau$ are computed based on the encoded task embedding vector $v$, which is further used to modulate the meta-learned prior parameters of the task network. The task network (introduced later at Section 4.2) can be an arbitrarily parameterized function, with multiple building blocks (or layers) such as deep convolutional networks [11], or multi-layer recurrent networks [32]. To modulate the parameters of each block in the task network as good initialization for solving the target task, we apply block-wise transformations to scale and shift the output activation of each hidden unit in the network (*i.e.* the output of a channel of a convolutional layer or a neuron of a fully-connected layer). Specifically, the modulation network produces the modulation vectors for each block $i$, denoted as

$$\tau_i = g_i(v; \omega_g), \text{where} \ \ i = 1, \cdots, N, \tag{2}$$

where $N$ is the number of blocks in the task network. We formalize the procedure of applying modulation as: $\phi_i = \theta_i \odot \tau_i$, where $\phi_i$ is the modulated prior parameters for the task network, and $\odot$ represents a general modulation operator. We investigate some representative modulation operations including attention-based (softmax) modulation [26, 47] and feature-wise linear modulation (FiLM) [31, 30, 13]. We empirically observe that FiLM performs better and more stable than attention-based modulation (see Section 5 for details), and therefore use FiLM as default operator for modulation. The details of these modulation operators can be found in the supplementary material.

## 4.2 Task Network

The parameters of each block of the task network are modulated using the task-specific parameters $\tau = \{\tau_i \mid i = 1, \cdots, N\}$ generated by the modulation network, which can generate a mode-aware initialization in the parameter space $f(x; \theta, \tau)$. After the modulation step, few steps of gradient descent are performed on the meta-learned prior parameters of the task network to further optimize

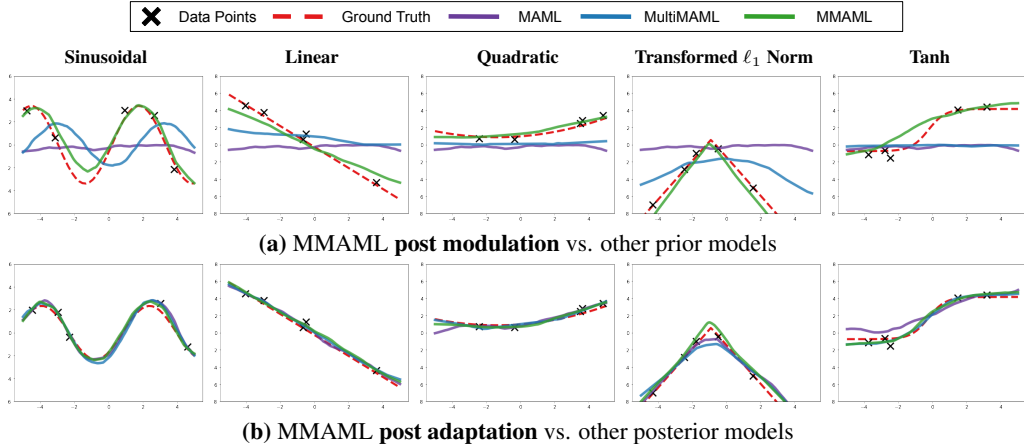

**(a)** MMAML **post modulation** vs. other prior models

**(b)** MMAML **post adaptation** vs. other posterior models

Figure 2: Qualitative Visualization of Regression on Five-modes Simple Functions Dataset. **(a)**: We compare the predicted function shapes of modulated MMAML against the prior models of MAML and Multi-MAML, before gradient updates. Our model can fit the target function with limited observations and no gradient updates. **(b)**: The predicted function shapes after five steps of gradient updates, MMAML is qualitatively better. More visualizations in Supplementary Material.

the objective function for a target task $\mathcal{T}_i$. Note that the task-specific parameters $\tau_i$ are kept fixed and only the meta-learned prior parameters of the task network are updated. We describe the concrete procedure in the form of the pseudo-code as shown in Algorithm 1. The same procedure of modulation and gradient-based optimization is used both during meta-training and meta-testing time.

Detailed network architectures and training hyper-parameters are different by the domain of applications, we defer the complete details to the supplementary material.

# 5   Experiments

We evaluate our method (MMAML) and baselines in a variety of domains including regression, image classification, and reinforcement learning, under the multimodal task distributions. We consider the following model-agnostic meta-learning baselines:

- **MAML [5]** represents the family of model-agnostic meta-learners. The architecture of MAML on each individual domain is designed to be the same as task network in MMAML.
- **Multi-MAML** consists of $M$ (the number of modes) MAML models and each of them is specifically trained on the tasks sampled from a single mode. The performance of this baseline is evaluated by choosing models based on *ground-truth task-mode labels*. This baseline can be viewed as the upper-bound of performance for MAML. If it outperforms MAML, it indicates that MAML's performance is degenerated due to the multimodality of task distributions. Note that directly comparing the other algorithms to Multi-MAML is not fair as it uses additional information which is not available in real world scenarios.

Note that we aim to develop a general model-agnostic meta-learning framework and therefore the comparison to methods that achieved great performance on only an individual domain are omitted. A more detailed discussion can be found in the supplementary material.

## 5.1   Regression Experiments

**Setups.** We experiment with our models in multimodal few-shot regression. In our setup, five pairs of input/output data $\{x_k, y_k\}_{k=1,...,K}$ are sampled from a one dimensional function and provided to a learning model. The model is asked to predict $L$ output values $y_1^q, ..., y_L^q$ for input queries $x_1^q, ..., x_L^q$. To construct the multimodal task distribution, we set up five different functions: sinusoidal, linear, quadratic, transformed $\ell_1$ norm, and hyperbolic tangent functions, and treat them as discrete task modes. We then evaluate three different task combinations with two functions, three functions and five functions in them. For each task, five pairs of data are sampled and Gaussian noise is added to the

Table 1: Mean square error (MSE) on the **multimodal 5-shot regression** with 2, 3, and 5 modes. A Gaussian noise with $\mu = 0$ and $\sigma = 0.3$ is applied. Multi-MAML uses ground-truth task modes to select the corresponding MAML model. Our method (with FiLM modulation) outperforms other methods by a margin.

| Method | 2 Modes | | 3 Modes | | 5 Modes | |
|---|---|---|---|---|---|---|
| | Post Modulation | Post Adaptation | Post Modulation | Post Adaptation | Post Modulation | Post Adaptation |
| MAML [5] | - | 1.085 | - | 1.231 | - | 1.668 |
| Multi-MAML | - | 0.433 | - | 0.713 | - | 1.082 |
| LSTM Learner | 0.362 | - | 0.548 | - | 0.898 | - |
| **Ours: MMAML (Softmax)** | 1.548 | 0.361 | 2.213 | **0.444** | 2.421 | 0.939 |
| **Ours: MMAML (FiLM)** | 2.421 | **0.336** | 1.923 | **0.444** | 2.166 | **0.868** |

Table 2: Classification testing accuracies on the **multimodal few-shot image classification** with 2, 3, and 5 modes. Multi-MAML uses ground-truth dataset labels to select corresponding MAML models. Our method outperforms MAML and achieve comparable results with Multi-MAML in all the scenarios.

| Method & Setup | 2 Modes | | | 3 Modes | | | 5 Modes | | |
|---|---|---|---|---|---|---|---|---|---|
| Way | 5-way | | 20-way | 5-way | | 20-way | 5-way | | 20-way |
| Shot | 1-shot | 5-shot | 1-shot | 1-shot | 5-shot | 1-shot | 1-shot | 5-shot | 1-shot |
| MAML [5] | 66.80% | 77.79% | 44.69% | 54.55% | 67.97% | 28.22% | 44.09% | 54.41% | 28.85% |
| Multi-MAML | 66.85% | 73.07% | **53.15%** | 55.90% | 62.20% | **39.77%** | 45.46% | 55.92% | 33.78% |
| MMAML (ours) | **69.93%** | **78.73%** | 47.80% | **57.47%** | **70.15%** | 36.27% | **49.06%** | **60.83%** | **33.97%** |

output value $y$, which further increases the difficulty of identifying which function generated the data. Please refer to the supplementary materials for details and parameters for regression experiments.

**Baselines and Our Approach.** As mentioned before, we have MAML and Multi-MAML as two baseline methods, both with MLP task networks. Our method (MMAML) augments the task network with a modulation network. We choose to use an LSTM to serve as the modulation network due to its nature as good at handling sequential inputs and generate predictive outputs. Data points (sorted by $x$ value) are first input to this network to generate task-specific parameters that modulate the task network. The modulated task network is then further adapted using gradient-based optimization. Two variants of modulation operators – softmax and FiLM are explored to be used in our approach. Additionally, to study the effectiveness of the LSTM model, we evaluate another baseline (referred to as the LSTM Learner) that uses the LSTM as the modulation network (with FiLM) but does not perform gradient-based updates. Please refer to the supplementary materials for concrete specification of each model.

**Results.** The quantitative results are shown in Table 1. We observe that MAML has the highest error in all settings and that incorporating task identity (Multi-MAML) can improve over MAML significantly. This suggests that MAML degenerates under multimodal task distributions. The LSTM learner outperforms both MAML and Multi-MAML, showing that the sequence model can effectively tackle this regression task. MMAML improves over the LSTM learner significantly, which indicates that with a better initialization (produced by the modulation network), gradient-based optimization can lead to superior performance. Finally, since FiLM outperforms Softmax consistently in the regression experiments, we use it for as the modulation method in the rest of experiments.

We visualize the predicted function shapes of MAML, Multi-MAML and MMAML (with FiLM) in Figure 2. We observe that modulation can significantly modify the prediction of the initial network to be close to the target function (see Figure 2 (a)). The prediction is then further improved by gradient-based optimization (see Figure 2 (b)). tSNE [23] visualization of the task embedding (Figure 3) shows that our embedding learns to separate the input data of different tasks, which can be seen as a evidence of the mode identification capability of MMAML.

## 5.2 Image Classification

**Setup.** The task of few-shot image classification considers the problem of classifying images into $N$ classes with a small number ($K$) of labeled samples available (*i.e.* $N$-way $K$-shot). To create a multimodal few-shot image classification task, we combine multiple widely used datasets (OMNIGLOT [17], MINI-IMAGENET [34], FC100 [29], CUB [50], and AIRCRAFT [24]) to form a

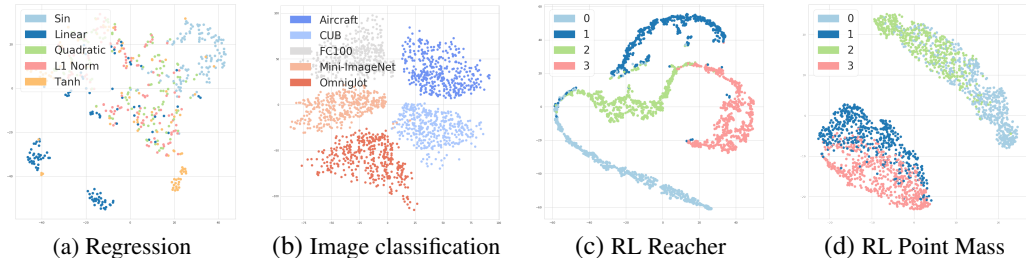

| (a) Regression | (b) Image classification | (c) RL Reacher | (d) RL Point Mass |

Figure 3: tSNE plots of the task embeddings produced by our model from randomly sampled tasks; marker color indicates different modes of a task distribution. The plots (b) and (d) reveal a clear clustering according to different task modes, which demonstrates that MMAML is able to identify the task from a few samples and produce a meaningful embedding $v$. (a) Regression: the distance between modes aligns with the intuition of the similarity of functions (*e.g.* a quadratic function can sometimes be similar to a sinusoidal or a linear function while a sinusoidal function is usually different from a linear function) (b) Few-shot image classification: each dataset (*i.e.* mode) forms its own cluster. (c-d) Reinforcement learning: The numbered clusters represent different modes of the task distribution. The tasks from different modes are clearly clustered together in the embedding space.

meta-dataset following the train/test splits used in the prior work, similar to [46]. The details of all the datasets can be found in the supplementary material.

We train models on the meta-datasets with two modes (OMNIGLOT and MINI-IMAGENET), three modes (OMNIGLOT, MINI-IMAGENET, and FC100), and five modes (all the five datasets). We use a 4-layer convolutional network for both MAML and our task network.

**Results.** The results are shown in Table 2, demonstrating that our method achieves better results against MAML and performs comparably to Multi-MAML. The performance gap between ours and MAML becomes larger when the number of modes increases, suggesting our method can handle multimodal task distributions better than MAML. Also, compared to Multi-MAML, our method achieves slightly better results partially because our method learns from all the datasets yet each Multi-MAML is likely to overfit to a single dataset with a smaller number of classes (*e.g.* MINI-IMAGENET and FC100). This finding aligns with the current trend of meta-learning from multiple datasets [46]. The detailed performance on each dataset can be found in the supplementary material.

To gain insights to the task embeddings $v$ produced by our model, we randomly sample 2000 5-mode 5-way 1-shot tasks and employ tSNE to visualize $v$ in Figure 3 (b), showing that our task embedding network captures the relationship among modes, where each dataset forms an individual cluster. This structure shows that our task encoder learns a reasonable task embedding space, which allows the modulation network to modulate the parameters of the task network accordingly.

## 5.3 Reinforcement Learning

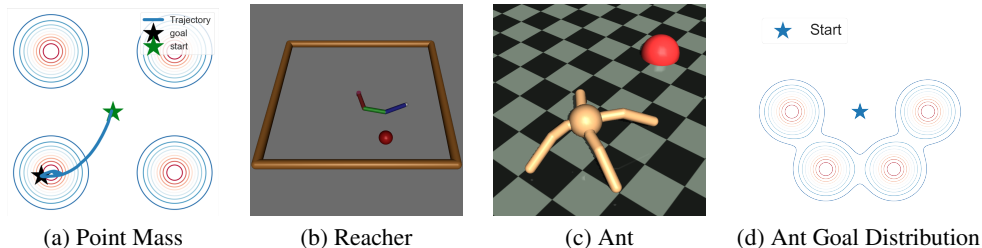

| (a) Point Mass | (b) Reacher | (c) Ant | (d) Ant Goal Distribution |

Figure 4: **RL environments**. Three environments are used to explore the capability of MMAML to adapt in multimodal task distributions in RL. In all of the environments the agent is tasked to reach a goal marked by a star of a sphere in the figures. The goals are sampled from a multimodal distribution in two or three dimensions depending on the environment. In POINT MASS (a) the agent navigates a simple point mass agent in 2-dimensions. In REACHER (b) the agent controls a 3-link robot arm in 2-dimensions. In ANT (c) the agent controls four-legged ant robot and has to navigate to the goal. The goals are sampled from a 2-dimensional distribution presented in figure (d), while the agent itself is 3-dimensional.

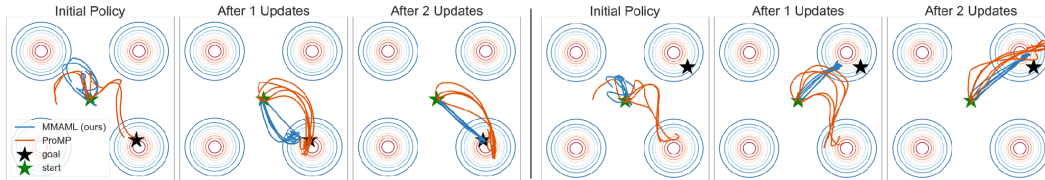

Figure 5: Visualizations of MMAML and ProMP trajectories in the 4-mode Point Mass 2D environment. Each trajectory originates in the green star. The contours present the multimodal goal distribution. Multiple trajectories are shown per each update step. For each column: **the leftmost figure** depicts the initial exploratory trajectories without modulation or gradient adaptation applied. **The middle figure** presents ProMP after one gradient adaptation step and MMAML after a gradient adaptation step and the modulation step, which are computed based on the same initial trajectories. **The figure on the right** presents the methods after two gradient adaptation steps in addition to the MMAML modulation step.

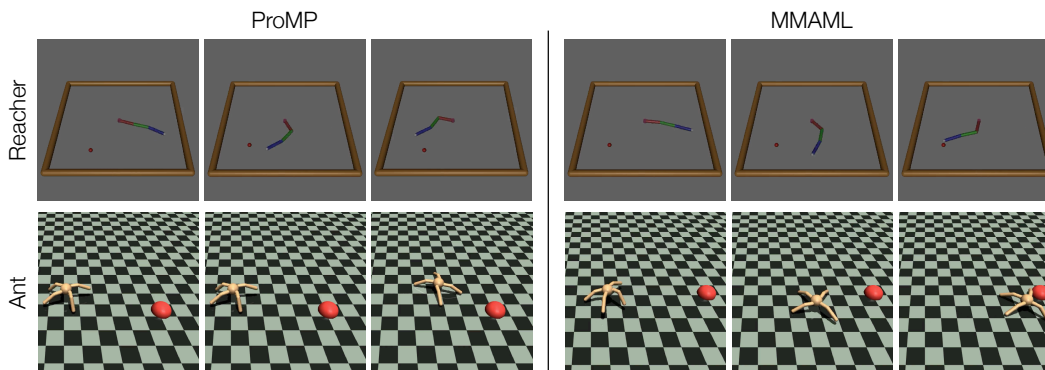

Figure 6: Visualizations of MMAML and ProMP trajectories in the ANT and REACHER environments. The figures represent randomly sampled trajectories after the modulation step and two gradient steps for REACHER and three for ANT. Each frame sequence represents a complete trajectory, with the beginning, middle and end of the trajectories captured by the left, middle and right frames respectively. Videos of the trained agents can be found at https://vuoristo.github.io/MMAML/.

**Setup.** Along with few-shot classification and regression, reinforcement learning (RL) has been a central problem where meta-learning has been studied [40, 39, 52, 5, 25, 35]. Similarly to the other domains, the objective in meta-reinforcement learning (meta-RL) is to adapt to a novel task based on limited experience with the task. For RL problems, the inner loop updates of gradient-based meta-learning take the form of policy gradient updates. For a more detailed description of the meta-RL problem setting, we refer the reader to [35].

We seek to verify the ability of MMAML to learn to adapt to tasks sampled from multimodal task distributions based on limited interaction with the environment. We do so by evaluating MMAML and the baselines on four continuous control environments using the MuJoCo physics simulator [45]. In all of the environments, the agent is rewarded on every time step for minimizing the distance to the goal. The goals are sampled from multimodal goal distributions with environment specific parameters. The agent does not observe the location of the goal directly but has to learn to find it based on the reward instead. To provide intuition on the environments, illustrations of the robots are presented in Figure 4. Examples of trajectories are presented in Figure 5 for POINT MASS and in Figure 6 for ANT and REACHER. Complete details of the environments and goal distributions can be found in the supplementary material.

**Baselines and Our Approach.** To identify the mode of a task distribution with MMAML, we run the initial policy to interact with the environment and collect a batch of trajectories. These initial trajectories are used for two purposes: computing the adapted parameters using a gradient-based update and modulating the updated parameters based on the task embedding $\upsilon$ computed by the modulation network. The modulation vectors $\tau$ are kept fixed for the subsequent gradient updates. Descriptions of the network architectures and training hyperparameters are deferred to

Table 3: The mean and standard deviation of cumulative reward per episode for multimodal reinforcement learning problems with 2, 4 and 6 modes reported across 3 random seeds. Multi-ProMP is ProMP trained on an easier task distribution which consists of a single mode of the multimodal distribution to provide an approximate upper limit on the performance on each task.

| Method | POINT MASS 2D | | | REACHER | | | ANT | |
|---|---|---|---|---|---|---|---|---|
| | 2 Modes | 4 Modes | 6 Modes | 2 Modes | 4 Modes | 6 Modes | 2 Modes | 4 Modes |
| ProMP [35] | -397 ± 20 | -523 ± 51 | -330 ± 10 | -12 ± 2.0 | -13.8 ± 2.5 | -14.9 ± 2.9 | -761 ± 48 | -953 ± 46 |
| Multi-ProMP | -109 ± 6 | -109 ± 6 | -92 ± 4 | -4.3 ± 0.1 | -4.3 ± 0.1 | -4.3 ± 0.1 | -624 ± 38 | -611 ± 31 |
| Ours | -136 ± 8 | -209 ± 32 | -169 ± 48 | -10.0 ± 1.0 | -11.0 ± 0.8 | -10.9 ± 1.1 | -711 ± 25 | -904 ± 37 |

the supplementary material. Due to credit-assignment problems present in the MAML for RL algorithm [5] as identified in [35], we optimize our policies and modulation networks with ProMP [35] algorithm, which resolves these issues.

We use ProMP both as the training algorithm for MMAML and as a baseline. Multi-ProMP is an artificial baseline to show the performance of training one policy for each mode using ProMP. In practice we train an agent for only one of the modes since the task distributions are symmetric and the agent is initialized to a random pose.

**Results.** The results for the meta-RL experiments presented in Table 3 show that MMAML consistently outperforms the unmodulated ProMP. The good performance of Multi-ProMP, which only considers a single mode suggests that the difficulty of adaptation in our environments results mainly from the multiple modes. We find that the difficulty of the RL tasks does not scale directly with the number of modes, *i.e.* the performance gap between MMAML and ProMP for POINT MASS with 6 modes is smaller than the gap between them for 4 modes. We hypothesize the more distinct the different modes of the task distribution are, the more difficult it is for a single policy initialization to master. Therefore, adding intermediate modes (going from 4 to 6 modes) can in some cases make the task distribution easier to learn.

The tSNE visualizations of embeddings of random tasks in the POINT MASS and REACHER domains are presented in Figure 3. The clearly clustered embedding space shows that the task encoder is capable of identifying the task mode and the good results MMAML achieves suggest that the modulation network effectively utilizes the task embeddings to tackle the multimodal task distribution.

## 6 Conclusion

We present a novel approach that is able to leverage the strengths of both model-based and model-agnostic meta-learners to discover and exploit the structure of multimodal task distributions. Given a few samples from a target task, our modulation network first identifies the mode of the task distribution and then modulates the meta-learned prior in a parameter space. Next, the gradient-based meta-learner efficiently adapts to the target task through gradient updates. We empirically observe that our modulation network is capable of effectively recognizing the task modes and producing embeddings that captures the structure of a multimodal task distribution. We evaluated our proposed model in multimodal few-shot regression, image classification and reinforcement learning, and achieved superior generalization performance on tasks sampled from multimodal task distributions.

## Acknowledgment

This work was initiated when Risto Vuorio was at SK T-Brain and was partially supported by SK T-Brain. The authors are grateful for the fruitful discussion with Kuan-Chieh Wang, Max Smith, and Youngwoon Lee. The authors appreciate the anonymous NeurIPS reviewers as well as the anonymous reviewers who rejected this paper but provided constructive feedback for improving this paper in previous submission cycles.

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
