[Supplementary Material]

# Supplementary Material of
# Multimodal Model-Agnostic Meta-Learning via Task-Aware Modulation

**Risto Vuorio**[*1]     **Shao-Hua Sun**[*2]     **Hexiang Hu**[2]     **Joseph J. Lim**[2]

[1]University of Michigan     [2]University of Southern California

vuoristo@gmail.com     {shaohuas, hexiangh, limjj}@usc.edu

## Contents

## A  Details on Modulation Operators

**Attention based modulation** has been widely used in modern deep learning models and has proved its effectiveness across various tasks [12, 24, 25, 27]. Inspired by the previous works, we employed attention to modulate the prior model. In concrete terms, attention over the outputs of all neurons (Softmax) or a binary gating value (Sigmoid) on each neuron's output is computed by the modulation network. These modulation vectors $\tau$ are then used to scale the pre-activation of each neural network layer $\mathbf{F}_\theta$, such that $\mathbf{F}_\phi = \mathbf{F}_\theta \otimes \tau$. Note that here $\otimes$ represents a channel-wise multiplication.

**Feature-wise linear modulation (FiLM)** has been proposed to modulate neural networks for achieving the conditioning effects of data from different modalities. We adopt FiLM as an option for modulating our task network parameters. Specifically, the modulation vectors $\tau$ are divided into two components $\tau_\gamma$ and $\tau_\beta$ such that for a certain layer of the neural network with its pre-activation $\mathbf{F}_\theta$, we would have $\mathbf{F}_\phi = \mathbf{F}_\theta \otimes \tau_\gamma + \tau_\beta$. It can be viewed as a more generic form of attention mechanism. Please refer to [16] for the complete details. In a recent few-shot image classification paper [13], FiLM modulation is used in a metric learning model and achieves high performance. Similarly, employing FiLM modulation has been shown effective on a variety of tasks such as image synthesis [1, 7, 8, 14], visual question answering [15, 16], style transfer [3], recognition [6, 23], reading comprehension [2], etc.

## B  Further Discussion on Related Works

**Discussions on Task-Specific Adaptation/Modulation.**  As mentioned in the related work of the main text, some recent works [11, 13, 26] leverage the task-specific adaptation or modulation to achieve few-shot image classification. Now we discuss about them in details. [13] propose to learn a task-specific network that adapts the weight of the visual embedding networks via feature-wise linear modulation (FiLM) [16]. Similarly, [26] learns to perform similar task-specific adaptation for few-shot image classification via Transformer [22]. [11] learns a visual embedding network with a task-specific metric and task-agnostic parameters, where the task-specific metric can be update via a fixed steps of gradient updates similar to [4]. In contrast, we aim to leverage the power of task-specific modulation to develop a more powerful model-agnostic meta-learning framework, which is able to effectively adapt to tasks sampled from a multimodal task distribution. Note that our proposed framework is capable of solving few-shot regression, classification, and reinforcement learning tasks.

## C  Baselines

Since we aim to develop a general model-agnostic meta-learning framework, the comparison to methods that achieved great performance on only an individual domain are omitted.

**Image Classification.**  While Prototypical networks [19], Proto-MAML [21], and TADAM [13] learn a metric space for comparing samples and therefore are not directly applicable to regression and reinforcement learning domains, we believe it would be informative to evaluate those methods on our multimodal image classification setting. For this purpose, we refer the readers to a recent work [21] which presents extensive experiments on a similar multimodal setting with a wide range of methods, including model-based (RNN-based) methods, model-agnostic meta-learners, and metric-based methods.

**Reinforcement Learning.**  We believe comparing MMAML to ProMP [18] on reinforcement learning tasks highlights the advantage of using a separate modulation network in addition to the task network, given that in the reinforcement learning setting MMAML uses ProMP as the optimization algorithm. Besides ProMP, Bayesian MAML [9] presents an appealing baseline for multimodal task distributions. We tried to run Bayesian MAML on our multimodal task distributions but had technical difficulties with it. The source code for Bayesian MAML in classification and regression is not publicly available.

|   |   |   |
|---|---|---|
| (a) 2-Mode Regression | (b) 3-Mode Regression | (c) 5-Mode Regression |

Figure 1: tSNE plots of the task embeddings produced by our model from randomly sampled tasks for regression. We choose to visualize the corresponding task embeddings of two modes, three modes and five modes.

# D    Additional Experimental Details

## D.1    Regression

### D.1.1    Setups

To form multimodal task distributions for regression, we consider a family of functions including sinusoidal functions (in forms of $A \cdot \sin w \cdot x + b + \epsilon$, with $A \in [0.1, 5.0]$, $w \in [0.5, 2.0]$ and $b \in [0, 2\pi]$), linear functions (in forms of $A \cdot x + b$, with $A \in [-3, 3]$ and $b \in [-3, 3]$), quadratic functions (in forms of $A \cdot (x - c)^2 + b$, with $A \in [-0.15, -0.02] \cup [0.02, 0.15]$, $c \in [-3.0, 3.0]$ and $b \in [-3.0, 3.0]$ ), $\ell_1$ norm function (in forms of $A \cdot |x - c| + b$, with $A \in [-0.15, -0.02] \cup [0.02, 0.15]$, $c \in [-3.0, 3.0]$ and $b \in [-3.0, 3.0]$), and hyperbolic tangent function (in forms of $A \cdot tanh(x - c) + b$, with $A \in [-3.0, 3.0]$, $c \in [-3.0, 3.0]$ and $b \in [-3.0, 3.0]$). Gaussian observation noise with $\mu = 0$ and $\epsilon = 0.3$ is added to each data point sampled from the target task. In all the experiments, $K$ is set to 5 and $L$ is set to 10. We report the mean squared error (MSE) as the evaluation criterion. Due to the multimodality and uncertainty, this setting is more challenging comparing to [5].

### D.1.2    Models and Optimization

In the regression task, we trained a 4-layer fully connected neural network with the hidden dimensions of 100 and ReLU non-linearity for each layer, as the base model for both MAML and MMAML. In MMAML, an additional model with a Bidirectional LSTM of hidden size 40 is trained to generate $\tau$ and to modulate each layer of the base model. We used the same hyper-parameter settings as the regression experiments presented in [5] and used Adam [10] as the meta-optimizer. For all our models, we train on 5 meta-train examples and evaluate on 10 meta-val examples to compute the loss.

### D.1.3    Evaluation Protocol

In the evaluation of regression experiments, we samples 25,000 tasks for each task mode and evaluate all models with 5 gradient steps during the adaptation (if applicable), with the adaptation learning rate set to be the one models learned with. Therefore, the results for 2 mode experiments is computed over 50,000 tasks, corresponding 3 mode experiment is computed over 75,000 tasks and 5 mode has 125,000 tasks in total. We evaluate all methods over the function range between -5 and 5, and report the accumulated mean squared error as performance measures.

### D.1.4    Effect of Modulation and Adaptation

We analyze the effect of modulation and adaptation steps on the regression experiments. Specifically, we show both the qualitative and quantitative results on the 5-mode regression task, and plot the induced function curves as well as measure the Mean Squared Error (MSE) after applying modulation step or both modulation and adaptation step. Note that MMAML starts from a learned prior parameters (denoted as *prior params*), and then sequentially performs modulation and adaptation steps. The results are shown in the Figure 2 and Table 1. We see that while inference with prior parameters itself induces high error, adding modulation as well as further adaptation can significantly reduce such

Table 2: Dataset details.

| Dataset | Train classes | Validation classes | Test classes | Image size | Image channel | Image content |
|---|---|---|---|---|---|---|
| OMNIGLOT | 4112 | 688 | 1692 | $28 \times 28$ | 1 | handwritten characters |
| MINI-IMAGENET | 64 | 16 | 20 | $84 \times 84$ | 3 | objects |
| FC100 | 64 | 16 | 20 | $32 \times 32$ | 3 | objects |
| CUB | 140 | 30 | 30 | $\sim 500 \times 500$ | 3 | birds |
| AIRCRAFT | 70 | 15 | 15 | $\sim$ 1-2 Mpixels | 3 | aircrafts |

error. We can see that the modulation step is trying to seek a rough solution that captures the shape of the target curve, and the gradient based adaptation step refines the induced curve.

Figure 2: 5-mode Regression: Visualization with Linear & Quadratic Function.

Table 1: 5-mode Regression: Performance measured in mean squared error (MSE).

| MMAML | MSE |
|---|---|
| Prior Params | 17.299 |
| + Modulation | 2.166 |
| + Adaptation | 0.868 |

## D.2 Image Classification

### D.2.1 Meta-dataset

To create a meta-dataset by merging multiple datasets, we utilize five popular datasets: OMNIGLOT, MINI-IMAGENET, FC100, CUB, and AIRCRAFT. The detailed information of all the datasets are summarized in Table 2. To fit the images from all the datasets to a model, we resize all the images to $84 \times 84$. The images randomly sampled from all the datasets are shown in Figure 3, demonstrating a diverse set of modes.

Figure 3: Examples of images from all the datasets.

### D.2.2 Hyperparameters

We present the hyperparameters for all the experiments in Table 3. We use the same set of hyperparameters to train our model and MAML for all experiments, except that we use a smaller meta batch-size for 20-way tasks and train the jobs for more iterations due to the limited memory of GPUs that we have access to.

We use 15 examples per class for evaluating the post-update meta-gradient for all the experiments, following [5, 17]. All the trainings use the Adam optimizer [10] with default hyperparameters.

Table 3: Hyperparameters for multimodal few-shot image classification experiments. We experiment different hyperparameters for each dataset for Multi-MAML. The dataset group **Grayscale** includes OMNIGLOT and **RGB** includes MINI-IMAGENET and FC100, CUB, and AIRCRAFT.

| Method | Setup | Dataset group | Slow lr | Fast lr | Meta bach-size | Number of updates | Training iterations |
|---|---|---|---|---|---|---|---|
| MAML | 5-way 1-shot | | | | | | |
| | 5-way 5-shot | | | | | | |
| | 5-way 1-shot | - | 0.001 | 0.05 | 10 | 5 | 60000 |
| MMAML (ours) | 5-way 5-shot | | | | | | |
| MAML | 20-way 1-shot | | | | | | |
| | 20-way 3-shot | | | | | | |
| | 20-way 1-shot | - | 0.001 | 0.05 | 5 | 5 | 80000 |
| MMAML (ours) | 20-way 3-shot | | | | | | |
| Multi-MAML | 5-way 1-shot | Grayscale | | 0.4 | 10 | 1 | |
| | | RGB | | 0.01 | 4 | 5 | 60000 |
| | 5-way 5-shot | Grayscale | | 0.4 | 10 | 1 | |
| | | RGB | 0.001 | 0.01 | 4 | 5 | |
| | 20-way 1-shot | Grayscale | | 0.1 | 4 | 5 | |
| | | RGB | | 0.01 | 2 | 5 | 80000 |
| | 20-way 3-shot | Grayscale | | 0.1 | 4 | 5 | |
| | | RGB | | 0.01 | 2 | 5 | |

Table 4: The performance (classification accuracy) on the **multimodal few-shot image classification** with **2 modes** on each dataset.

| Setup | Method | Datasets | | |
|---|---|---|---|---|
| | | OMNIGLOT | MINI-IMAGENET | OVERALL |
| 5-way 1-shot | MAML | 89.24% | 44.36% | 66.80% |
| | Multi-MAML | 97.78% | 35.91% | 66.85% |
| | MMAML (ours) | 94.90% | 44.95% | 69.93% |
| 5-way 5-shot | MAML | 96.24% | 59.35% | 77.79% |
| | Multi-MAML | 98.48% | 47.67% | 73.07% |
| | MMAML (ours) | 98.47% | 59.00% | 78.73% |
| 20-way 1-shot | MAML | 55.36% | 15.67% | 35.52% |
| | Multi-MAML | 91.59% | 14.71% | 53.15% |
| | MMAML (ours) | 83.14% | 12.47% | 47.80% |

For Multi-MAML, since we train a MAML model for each dataset, it gives us the freedom to use different sets of hyperparameters for different datasets We tried our best to find the best hyperparameters for each dataset.

### D.2.3 Network Architectures

**Task Network.** For the task network, we use the exactly same architecture as the MAML convolutional network proposed in [5]. It consists of four convolutional layers with the channel size 32, 64, 128, and 256, respectively. All the convolutional layers have a kernel size of 3 and stride of 2. A batch normalization layer follows each convolutional layer, followed by ReLU. With the input tensor size of $(n \cdot k) \times 84 \times 84 \times 3$ for a $n$-way $k$-shot task, the output feature maps after the final convolutional layer have a size of $(n \cdot k) \times 6 \times 6 \times 256$. The feature maps are then average pooled along spatial dimensions, resulting feature vectors with a size of $(n \cdot k) \times 256$. A linear fully-connected layer takes the feature vector as input, and produce a classification prediction with a size of $n$ for n-way classification tasks.

**Task Encoder.** For the task encoder, we use the exactly same architecture as the task network. It consists of four convolutional layers with the channel size 32, 64, 128, and 256, respectively. All the convolutional layers have a kernel size of 3, stride of 2, and use valid padding. A batch normalization layer follows each convolutional layer, followed by ReLU. With the input tensor size of $(n \cdot k) \times 84 \times 84 \times 3$ for a $n$-way $k$-shot task, the output feature maps after the final convolutional layer have a size of $(n \cdot k) \times 6 \times 6 \times 256$. The feature maps are then average pooled along spatial dimensions, resulting feature vectors with a size of $(n \cdot k) \times 256$. To produce an aggregated embedding vector from all the feature vectors representing all samples, we perform an average pooling, resulting a feature vector with a size of 256. Finally, a fully-connected layer followed by ReLU takes the feature vector as input, and produce a task embedding vector $\upsilon$ with a size of 128.

Table 5: The performance (classification accuracy) on the **multimodal few-shot image classification** with **3 modes** on each dataset.

| Setup | Method | Datasets | | | |
|---|---|---|---|---|---|
| | | OMNIGLOT | MINI-IMAGENET | FC100 | OVERALL |
| 5-way 1-shot | MAML | 86.76% | 43.27% | 33.29% | 54.55% |
| | Multi-MAML | 97.78% | 35.91% | 34.00% | 55.90% |
| | MMAML (ours) | 93.67% | 41.07% | 33.67% | 57.47% |
| 5-way 5-shot | MAML | 95.11% | 61.48% | 47.33% | 67.97% |
| | Multi-MAML | 98.48% | 47.67% | 40.44% | 62.20% |
| | MMAML (ours) | 99.56% | 60.67% | 50.22% | 70.15% |
| 20-way 1-shot | MAML | 57.87% | 15.06% | 11.74% | 28.22% |
| | Multi-MAML | 91.59% | 14.71% | 13.00% | 39.77% |
| | MMAML (ours) | 85.00% | 13.00% | 10.81% | 36.27% |

Table 6: The performance (classification accuracy) on the **multimodal few-shot image classification** with **5 modes** on each dataset.

| Setup | Method | Datasets | | | | | |
|---|---|---|---|---|---|---|---|
| | | OMNIGLOT | MINI-IMAGENET | FC100 | CUB | AIRCRAFT | OVERALL |
| 5-way 1-shot | MAML | 83.63% | 37.78% | 33.70% | 86.96% | 36.74% | 35.48% |
| | Multi-MAML | 97.78% | 35.91% | 34.00% | 93.44% | 32.03% | 27.59% |
| | MMAML (ours) | 91.48% | 42.89% | 32.59% | 93.56% | 38.30% | 36.82% |
| 5-way 5-shot | MAML | 89.41% | 51.26% | 43.41% | 82.30% | 45.80% | 43.92% |
| | Multi-MAML | 98.48% | 47.67% | 40.44% | 98.56% | 45.70% | 47.29% |
| | MMAML (ours) | 97.96% | 51.29% | 44.08% | 97.88% | 53.80% | 51.53% |
| 20-way 1-shot | MAML | 59.10% | 15.49% | 11.75% | 59.45% | 16.31% | 31.57% |
| | Multi-MAML | 91.59% | 14.71% | 13.00% | 85.46% | 18.87% | 30.72% |
| | MMAML (ours) | 86.28% | 14.35% | 11.59% | 91.86% | 24.05% | 30.89% |

**Modulation MLPs** . Since the task network consists of four convolutional layers with the channel size 32, 64, 128, and 256 and modulating each of them requires producing both $\tau_\gamma$ and $\tau_\beta$, we employ four linear fully-connected layers to convert the task embedding vector $\upsilon$ to $\{\tau_{\gamma_1}, \tau_{\beta_1}\}$ (with a dimension of 32), $\{\tau_{\gamma_2}, \tau_{\beta_2}\}$ (with a dimension of 64), $\{\tau_{\gamma_3}, \tau_{\beta_3}\}$ (with a dimension of 128), and $\{\tau_{\gamma_4}, \tau_{\beta_4}\}$ (with a dimension of 256). Note the modulation for each layer is performed by $\theta_i \odot \gamma_i + \beta_i$, where $\odot$ denotes the Hadamard product.

## D.3 Reinforcement Learning

### D.3.1 Environments

The training curves for all environments are presented in Figure 5.

**POINT MASS** . We consider three variants of the POINT MASS environment with 2, 4, and 6 modes. The agent controls a point mass by outputting changes to the velocity. At every time step the agent receives the negative euclidean distance to the goal as the reward. The goals are sampled from a multimodal goal distribution by first selecting the mode center and then adding Gaussian noise to the goal location. In the 4 mode variant the modes are the points $(-5, -5), (-5, 5), (5, -5), (5, 5)$. In the 2 mode variant the modes are the points $(-5, -5), (5, 5)$. In the 6 mode variant the modes are the vertices of a regular hexagon with at distance 5 from the origin. All variants have noise scale of 2.0. Visualizations of agent trajectories can be found in Figure 7.

**REACHER** . We consider three variants of the REACHER environment with 2, 4, and 6 modes. The agent controls a 2-dimensional robot arm with three links simulated in the MuJoCo [20] simulator. The goal distribution is similar to the goal distributions in POINT MASS but different parameters are used to match the scale of the environment. The reward for the environment is

$$R(s, a) = -1 * (x_{point} - x_{goal})^2 - \|a\|^2$$

where $x_{point}$ is the location of the point of the arm, $x_{goal}$ if the location of the goal and $a$ is the action chosen by the agent. The modes of the goal distribution in the 4 mode variant are located at $(-0.225, -0.225), (0.225, -0.225), (-0.225, 0.225), (0.225, 0.225)$ and the goal noise has scale of 0.1. In the 2 mode variant the modes are located at $(-0.225, -0.225), (0.225, 0.225)$ and the noise

(a) 2-mode classification     (b) 3-mode classification     (c) 5-mode classification

Figure 4: tSNE plots of task embeddings produced in multimodal few-shot image classification domain. (a) 2-mode 5-way 1-shot (b) 3-mode 5-way 1-shot (c) 5-mode 5-way 5-shot.

(a) POINT MASS 2 Modes     (b) POINT MASS 4 Modes     (c) POINT MASS 6 Modes

(a) REACHER 2 Modes     (b) REACHER 4 Modes     (c) REACHER 6 Modes

(a) ANT 2 Modes     (a) ANT 4 Modes

Figure 5: Training curves for MMAML and ProMP in reinforcement learning environments. The curves indicate the average return per episode after gradient-based updates and modulation. The shaded region indicates standard deviation across three random seeds. The curves have been smoothed by averaging the values within a window of 10 steps.

scale is $0.1$. In the 6 mode variant the mode centers are the vertices of a regular hexagon with distance to the origin of $0.318$ and the noise scale is $0.1$.

**ANT** . We consider two variants of the ANT environment with two and four modes. The agent controls an ant robot with four limbs simulated in the MuJoCo [20] simulator. The reward for the environment is

$$R(s, a) = -1 * (x_{torso} - x_{goal})^2 - \lambda_{control} * \|a\|^2$$

where $x_{torso}$ is the location of the torso of the robot, $x_{goal}$ if the location of the goal, $\lambda_{control} = 0.1$ is the weighting for the control cost and $a$ is the action chosen by the agent. The modes of the goal distribution in the 4 mode variant are located at $(-4, 0)$, $(-2, 3.46)$, $(2, 3.46)$, $(4.0, 0)$ and the goal noise has scale of $0.8$. In the 2 mode variant the modes are located at $(-4.0, 0)$, $(4.0, 0)$ and the noise scale is $0.8$.

### D.3.2 Network Architectures and Hyperparameters

For all RL experiments we use a policy network with two 64-unit hidden layers. The modulation network in RL tasks consists of a GRU-cell and post processing layers. The inputs to the GRU are

Table 7: Hyperparameter settings for reinforcement learning.

| Environment | Algorithm | Training Iterations | Trajectory Length | Slow lr | Fast lr | Inner Gradient Steps | Clip eps |
|---|---|---|---|---|---|---|---|
| POINT MASS | MMAML ProMP Multi-ProMP | 400 | 100 | 0.0005 | 0.01 | 2 | 0.1 |
| REACHER | MMAML ProMP Multi-ProMP | 800 | 50 | 0.001 | 0.1 | 2 | 0.1 |
| ANT | MMAML ProMP Multi-ProMP | 800 | 250 | 0.001 | 0.1 | 3 | 0.1 |

the concatenated observations, actions and reward for each trajectory. The trajectories are processed separately. An MLP is used to process the last hidden states of each trajectory. The outputs of the MLPs are averaged and used by another MLP to compute the modulation vectors $\tau$. All MLPs have a single hidden layer of size 64.

We sample 40 tasks for each update step. For each gradient step for each task we sample 20 trajectories. The hyperparameters, which differ from setting to setting are presented in Table 7.

# E  Additional Experimental Results

## E.1  Regression

We show visualization of embeddings for regression experiments with a varying number of task modes as Figure 1. We observe a linear separation in the two task modes and three task modes scenarios, which indicates that our method is capable of identifying data from different task modes. On the visualization of five task mode, we observe that data from linear, transformed $\ell_1$ norm and hyperbolic tangent functions cluttered. This is due to the fact that those functions are very similar to each other, especially with the Gaussian noise we added in the output space.

## E.2  Image Classification

We provide the detailed performance of our method and the baselines on each individual dataset for all 2, 3, and 5 mode experiments, shown in Table 4, Table 5, and Table 6, respectively. Note that the main paper presents the overall performance (the last columns of each table) on each of 2, 3, and 5 mode experiments.

We found the results on OMNIGLOT and MINI-IMAGENET demonstrate similar tendency shown in [21]. Note that the performance of OMNIGLOT and FC100 might be slightly different from the results reported in the related papers because (1) all the images are resized and tiled along the spatial dimensions, (2) different hyperparamters are used, and (3) different numbers of training iterations.

Additional tSNE plots for predicted task embeddings of 2-mode 5-way 1-shot classification, 3-mode 5-way 1-shot classification, and 5-mode 20-way 1-shot classification are shown in Figure 4.

## E.3  Reinforcement Learning

Additional trajectories sampled from the 2D navigation environment are presented in Figure 7.

Figure 6: Additional qualitative results of the regression tasks. MMAML **after adaptation** vs. other posterior models.

Figure 7: Additional trajectories sampled from the point mass environment with MMAML and ProMP for six tasks. The contour plots represents the multimodal task distribution. The stars mark the start and goal locations. The curves depict five trajectories sampled using each method after zero, one and two update steps. In the figure, the modulation step takes place between the initial policy and the step after one update.