[Reviews · NeurIPS 2019]

Reviewer 1



Overall, the proposed method is original (it harness the linear modulation idea from the visual question answering field and the style transfer field) and efficient. The paper is well written and clear. The scope of the algorithm is large, making thus the proposed method useful and significant. The algorithms of the MAML family (Finn, C., Abbeel, P., & Levine, S. Model-agnostic meta-learning for fast adaptation of deep networks. 2017) are designed to find model weights that are a good starting fine-tuning point for a task not seen during training. To achieve this, these algorithms rely on a set of different tasks, a task being a set of annotated data and an error criterion. The model will be trained in such a way that a few steps of fine-tuning on these different tasks yield the lowest possible error on these tasks. Because several tasks were seen during training, it provides an efficient initialization for fine-tuning on a new test set task. Those frameworks yield great performance on several family of tasks, such as few-shot learning or reinforcement learning. MAML (and related) seeks a common initialization for all the tasks at hand, no matter how different they are. This paper stated that seeking a single initialization for an entire tasks distribution is limiting the achievable performance over this distribution and will thus prevent the algorithm to work on a diverse tasks distribution. Then, the paper proposes a new meta-learning method able to overcome this limitation, by having au auxiliary network modulating the initialization depending on the task mode and evaluates its effectiveness on several family of tasks. The proposed method is state-of-the-art on all the evaluated tasks. The Introduction and the Related Works sections of the paper deal with meta-learning in general and its limitations. There is no missing major works in the bibliography. The Related Works and the Preliminaries are focused on the MAML algorithm, which is normal because the current algorithm is built upon MAML and is fairly different from the other kinds of meta-learning methods. The method is well explained. Reading the supplementary materials may be required to understand the details of the modulation of the parameters. The FiLM modulation operation is taken from a paper in the visual questions answering field, but the field of style transfer has also used similar methods (AdaIN) to control the style of the output image based on the style of an input image. The figures are clear, but some of them (Figure 2 and 3) absolutely needs to be viewed in color. The Experiments section follows the same architecture as the MAML experiments : regression, few-shot classification, reinforcement learning. The experiments are adapted for the method but do not look like hand-crafted for this particular algorithm. For instance, merging datasets (with the same labels) seems a reasonable strategy to creates a multi-modal tasks distribution. The section is covering all the aspects of the conducted experiments, and many more details are available from the supplementary materials (such as the used networks’ architectures or the training hyper-parameters). The comparison baselines are well chosen, even if the MAML method has gotten a few general improvements. The source code is released along the paper, and is of great quality.

Reviewer 2



This paper suggests an extension to MAML that focuses on integrating model-based meta-learners and gradient-based meta learners. The method recognizes the task "mode" and adapts an initialization that can be learned through a few gradient steps. A practical optimization algorithm is presented for learning the proposed framework. The paper is technically sound. MMAML is tested on three different tasks, including regression on synthetic data, image classification on challenging datasets, and RL on standard datasets. In the image classification tests, for the two-mode case mini-imagenet has been used, while for the 3-mode case and 5-mode case, the additional datasets are not very challenging. The paper claims that the gap between the proposed method and MAML is larger when there are more modes, suggesting that the impact of the method is better seen in cases with more modes. However, it is not clear whether the improvement in performance is due to the ability to better handle higher number of modes or it is because the additional modes are simple tasks and don't have much different distributions. The Two-digit MNIST and three-digit MNIST datasets added on the 5-mode case are both easy tasks and have relatively similar distributions. While it is expected to have lower accuracies when more modes are present, perhaps because MNIST-based datasets are too easy, the five-mode experiments have very high accuracies as compared to 2-mode tests. Some other classification datasets such as CIFAR might be a better choice to evaluation. The paper is generally well-written and structured clearly. The idea of the paper has a good practical impact and solves a limitation of MAML. Update after authors feedback: Authors have addressed the concerns. I would like to update my overall score to 8.

Reviewer 3



This paper studies meta-learning in the context of a multi-modal task distribution (e.g. few-shot image classification where input-output pairs come from entirely different datasets). The authors first note that MAML—which finds a single initialization of the parameters and updates those parameters to the task via standard gradient updates— is not well-suited to this setup because the diversity of the tasks likely requires substantially different parameters. Motivated by this observation, this paper proposes an extension to MAML, called multi-modal MAML (MMAML), which is designed to capture the multi-modality in the parameter space. More specifically, this paper uses a separate modulation network that adapts the task parameters through an affine feature transformation. The modulated task parameters then undergo the usual MAML gradient update. The paper evaluates the proposed extension on three tasks: a synthetic few-shot regression problem, a few-shot image classification problem, and a meta-RL problem. They compare their method against two baselines: MAML and multi-MAML, a variant of MAML which has access to the ground-truth task mode label. For the regression task, they show that 1) MMAML significantly outperforms MAML (especially when the number of modes increases), and that 2) surprisingly MMAML also outperforms multi-MAML, and that 2) the modulation step is doing the most of the work, already resulting in decent performance on the regression tasks without using the gradient steps. More or less similar findings are reported for the image classification problem. Strengths: 1. The paper is well-written and the core idea is well-motivated and easy-to-follow 2. The method is quite versatile and is evaluated on a diverse set of tasks (ranging from synthetic regression to image classification and reinforcement learning) 3. I believe the baselines are sensible (though the paper would benefit from comparison to stronger methods, see weaknesses) Weaknesses: 1. I believe the paper would be much stronger if it compares against stronger baselines like prototypical nets, proto-MAML, bayesian MAML and TADAM. Though the authors mention these methods in the related work, they do not directly compare against them. In the supplementary material, they say the following about bayesian MAML: “We believe the model distribution is still unimodal (with a Gaussian prior), which is not well-designed to address multimodal task distributions (similar to MAML).” Such claims should be backed by empirical evaluations. 2. Although the authors show through t-SNE visualizations that the modulation network successfully separates the modes of the task distribution, I believe more can be done to investigate the learned modulation step. For example, one of the premises of the proposed method is that the pure gradient step of MAML can’t bring you far enough in parameter space to obtain good performance across nodes. I think it will be informative if you check the norm of the modulation step vs the norm of the gradient updates, and compare this to the norm of MAML updates. Typos: L83. Those UPDATE: After reading the rebuttal I have updated my score to 7.

[Author Response · NeurIPS 2019]

# 1 Author Response

We thank the reviewers for their valuable feedback. We will address the comments and the concerns as follows.

**Multi-MAML vs. MMAML (R1).** (1) As a whole, Multi-MAML uses all the training tasks from different modes but each of its $M$ (the number of modes) MAML models gets trained with only the tasks sampled from the corresponding mode. MMAML does not use more data. (2) Multi-MAML assumes tasks from different modes are unrelated, whereas MMAML does not have this assumption. Therefore, we conjecture that utilizing the similarity among tasks from different modes contributes to the superior performance of MMAML. (3) As R1 suggests, there are ways to improve Multi-MAML (*e.g.* share an encoder and select dedicated classifiers for different task modes). However, Multi-MAML is not practical as it requires ground truth mode labels. Therefore, it mainly serves the purpose of verifying whether the performance of MAML degrades in multimodal task distributions. We will clarify all these points in the revised paper.

**Formal Definitions (R1).** Thanks for the suggestion. We agree that including formal definitions of *task*, *task distribution*, and *multimodal task distribution* would make the paper clearer and will add them in the revised paper.

**Regression Training and Testing Tasks (R1).** The training and testing tasks of regression experiments are sampled from the same family of functions, which is a standard setup in the existing few-shot regression literature.

**Explanation on the Figure 3(a) (R1).** The regression clusters overlap (*i.e.* does not reveal a clear clustering) because the observed data are noisy (with Gaussian noise) and it can be difficult to infer task modes. For example, Figure 2 of the main paper shows that the observed data of the quadratic function looks similar to a linear function. More examples of functions with ambiguity over the mode can be found in the supplementary material.

**Classification Datasets (R2).** (1) *CIFAR*: FC100 dataset in the paper refers to a few-shot learning version of CIFAR100 dataset (introduced in TADAM). We will clarify it in the revision. (2) *More datasets*: as suggested by R2, we replaced the two MNIST datasets with two popular datasets (CUB-200-2011 [1] and Aircraft [2]) for 5-mode experiments, similar to [3]. The results in Table R1 show that MMAML outperforms the baselines, which is consistent with finding of our main paper. We will add this to the revised paper and provide additional analysis.

**Baselines (R3).** We originally aimed to compare our method to the baselines that are applicable to all regression, classification, and RL. Prototypical networks, Proto-MAML, and TADAM learn a metric space for comparing samples, which are not directly applicable to regression and RL. However, we agree that it would be informative to evaluate those methods on our multimodal classification setting. We will incorporate them into the revised paper.

The code for Bayesian MAML had not been made publicly available until the paper submission deadline. During the rebuttal period, we have tried to tune and run it but have not gotten meaningful results yet. We will further consult with the authors in order to get it working so that we can add it to the revised paper.

**Modulation and Adaptation (R3).** We agree that analyzing the effect of modulation and adaptation steps would be helpful. However, the norm of the modulation step ($\tau$) and the norm of the adaptation step (in the parameter space: $\theta$) are not directly comparable. As an alternative, we provide an additional analysis showing the effects of modulation and adaptation qualitatively (shown in Figure R1) and quantitatively (shown in Table R2) by testing a trained MMAML in the 5-mode regression task and measuring the MSE of each step. Note that MMAML starts from a learned prior parameters (denoted as *prior params*), and then performs modulation and adaptation steps.

Table R1: 5-mode Classification: Omniglot, Mini-ImageNet, FC100, CUB, and Aircraft.

| Setup | 5w1s | 5w5s | 20w1s |
|---|---|---|---|
| MAML | 44.09% | 54.41% | 28.85% |
| Multi-MAML | 45.46% | 55.92% | 33.78% |
| MMAML (ours) | **49.06%** | **60.83%** | **33.97%** |

Figure R1: 5-mode Regression: Visualization with Linear & Quadratic Function.

Table R2: 5-mode Regression: Performance measured in mean squared error (MSE).

| MMAML | MSE |
|---|---|
| Prior Params | 17.299 |
| **+ Modulation** | 2.166 |
| **+ Adaptation** | 0.868 |

# References

[1] C. Wah, S. Branson, P. Welinder, P. Perona, and S. Belongie. The caltech-ucsd birds-200-2011 dataset. 1

[2] S. Maji, J. Kannala, E. Rahtu, M. Blaschko, and A. Vedaldi. Fine-grained visual classification of aircraft. *arXiv preprint airxiv:1306.5151*. 1

[3] E. Triantafillou, T. Zhu, V. Dumoulin, P. Lamblin, K. Xu, R. Goroshin, C. Gelada, K. Swersky, P.-A. Manzagol, and H. Larochelle. Meta-dataset: A dataset of datasets for learning to learn from few examples. In *Meta-Learning Workshop at NeurIPS*, 2018. 1


[Meta-Review · NeurIPS 2019]

All three reviewers were satisfied with the authors' feedback and maintained their positive appreciation on this submission. Please note that reviewers are expecting/trusting that changes you committed to do will appear in the final version of the paper.